**Perspective**

# Plastics matter in the food system

Joe Yates [1] ✉, Megan Deeney[1], Jane Muncke [2], Bethanie Carney Almroth [3], Marie-France Dignac[4], Arturo Castillo Castillo[5], Winnie Courtene-Jones [6], Suneetha Kadiyala[1], Eva Kumar [7], Peter Stoett[8], Mengjiao Wang [9] & Trisia Farrelly [10]

Agriculture and food systems are major sources of plastic pollution but they are also vulnerable to their diverse lifecycle impacts. However, this problem is not well-recognized in global policy and scientific discourse, agendas, and monitoring of food systems. The United Nations-led Global Plastics Treaty, which has been under negotiation since 2022, is a critical opportunity to address pollution across the entire plastics lifecycle for more sustainable and resilient food systems. Here, we offer aspirational indicators for future monitoring of food systems' plastics related to (1) plastic polymers and chemicals, (2) land use, (3) trade and waste, and (4) environmental and human health. We call for interdisciplinary research collaborations to continue improving and harmonising the evidence base necessary to track and trace plastics and plastic chemicals in food systems. We also highlight the need for collaboration across disciplines and sectors to tackle this urgent challenge for biodiversity, climate change, food security and nutrition, health and human rights at a whole systems level.

## Plastic applications across food systems

In 2022, the United Nations Environment Assembly adopted a resolution to develop an international legally binding instrument on plastic pollution, known as The Global Plastics Treaty. This is an historic opportunity to comprehensively address the full life cycle of plastics, including production, design, use and disposal[1].

From farm to fork and back to farms, food systems are major contributors of plastics pollution, driven by an array of applications. Two examples—food production and packaging—illustrate this relationship:

Agriculture, fisheries, and aquaculture utilise an estimated 3.5% of global plastics, with diverse uses including polytunnels, mulches, feeding equipment, nets, encapsulated fertilisers and seeds, irrigation, and storage equipment[2,3]. To give a sense of scale, an estimated 13% of China's cultivated land mass is covered in plastic film mulch[4], while Spain's "Mar de Plástico" greenhouses and polytunnels are visible from space. Some of these agricultural plastics may provide critical functions for productivity and efficiency by controlling pests, mediating resource use, preventing water stress, and reducing spoilage, thus extending growing seasons and contributing to the supply and availability of fresh foods[3]. For farmers and communities in precarious economic, environmental and humanitarian circumstances these functions have been fundamental to economic viability and may partially explain their rapid expansion across global landscapes[5]. Beyond the farm gate, the $400-500 billion annual food and drink packaging industry represents an estimated 10-20% of all plastics ever produced[6]. Some of these plastics support the long supply chains of the modern food system by preserving food, extending transportation time and shelf life, and enabling the mass production and long-range distribution of foodstuffs.

However, these long, complex and often vertically integrated supply chains are recognised as part of an increasingly fragile food system that is failing to deliver healthy and sustainable diets[7]. A predominant focus on short-term productivity, efficiencies, and financial returns associated with various food system plastic applications, including those used in the supply of healthier foods, has obscured the externalities and true costs of these diverse materials. This has opened the door to oversimplifications and generalisations employed by petrochemical and plastic industries who claim unequivocally that their products support food security, ensure food safety and prevent food waste[8]. These claims simultaneously justify escalating production of, and reliance on single-use plastics.

Expanding our appraisal to encompass entire lifecycle indicators relevant to environmental, economic, and social sustainability and human health presents a more comprehensive picture, revealing that many food-related plastics may pose medium- to long-term threats to people and planet. Only by understanding these trade-offs across various dimensions can we design policies and programmes that account for regrettable consequences.

[1]Faculty of Epidemiology and Population Health, London School of Hygiene & Tropical Medicine (LSHTM), London, UK. [2]Food Packaging Forum Foundation, Zurich, Switzerland. [3]Department of Biological & Environmental Sciences, University of Gothenburg, Gothenburg, Sweden. [4]French National Institute for Agriculture, Food, and Environment (INRAE), Paris, France. [5]Faculty of Geosciences, Utrecht University, Utrecht, Netherlands. [6]School of Ocean Science, Bangor University, Menai Bridge, UK. [7]Independent Scientist, Helsinki, Finland. [8]Faculty of Social Science and Humanities, University of Ontario Institute of Technology, Oshawa, Canada. [9]Greenpeace Research Laboratories, Bioscience, University of Exeter, Exeter, UK. [10]Transdisciplinary Science Group, Cawthron Institute, Nelson, New Zealand. ✉e-mail: Joe.Yates@lshtm.ac.uk

## The entire plastics lifecycle

Even before reaching their intended applications across food systems, the true costs of plastics begin mounting. More than 98% of plastic polymers are derived from fossil fuels, requiring chemical and energy-intensive processes that produce greenhouse gas (GHG) emissions and other pollutants[9]. These processes directly, indirectly, and disproportionately expose frontline, fenceline and other vulnerable communities, including Indigenous Peoples, to toxic emissions to air, water, and soil.

Like all plastics, those used across food systems contain thousands of chemicals. At least 4219 of these chemicals are of concern, associated with hazardous properties like persistence, bioaccumulation, mobility and toxicity[10], and/or are known to disrupt the endocrine systems of humans and other animals. Large numbers of plastic chemicals have not been assessed at all, while many deemed safe have not been transparently tested against the latest hazards-based safety criteria. Plastic chemicals migrate into foodstuffs, contributing to a broad range of adverse health outcomes[11,12].

An estimated two thirds of all plastics are short-lived and the vast majority of plastic food and beverage packaging is single use[13,14]. During and after their intended applications, plastics and their associated chemicals contaminate the environment at staggering rates. The vast quantities of plastics used in agriculture, horticulture, aquaculture and fisheries are utilised for less than one year, poorly managed, lost or discarded, constituting major sources of macro, micro and nano plastic (MNP) pollution[2]. While plastic pollution of marine environments has gained widespread attention, the situation in agricultural soils may be worse[2], with evidence indicating significantly more MNPs in soils on farms using agricultural plastics than in those that do not[15]. The use of biosolids for fertilisation which contain anthropogenic MNPs and plastic chemicals (e.g. from washing of textiles or post-consumer waste) also contribute to the contamination of soils[16,17]. MNP contamination, by both conventional and bio-based/biodegradable plastics, is shown to fundamentally undermine soil structure, biodiversity, element cycling and carbon sequestration, presenting potentially profound medium to long-term implications for food production[2,18]. Furthermore, MNPs and plastic chemicals are shown to affect plant growth and are taken up by crops[19], representing a critical pathway into human diets.

Whether from agricultural production, food processing or food packaging, ever growing volumes of MNPs inevitably accumulate in foodstuffs and the environment where they are ingested and inhaled by humans. MNPs have been found in human tissues, blood, placentas and reproductive organs. The full human health consequences of this exposure is yet unknown, but emerging evidence paints a concerning picture[12].

Plastics were never designed to be recovered[20] and food system plastics are among the most challenging to retrieve due to spoilage and single-use design (i.e. packaging) and disintegration during use (i.e. agricultural mulches). Less than 10% of plastics are estimated to have been recycled[21], the remaining landfilled, incinerated or polluting nature. Recycling itself is resource- and often chemically- intensive, pollution emitting, and (re) introduces further MNPs and toxicants[22]. Plastics combined from different manufacturing sources are difficult to ascertain and control. As such, recycled plastics, including food packaging, are unpredictable and equally hazardous, if not more, than virgin plastics[23] thus undermining aspirations for healthy and sustainable circular food systems. Given the high capital costs of waste management infrastructure and increasing plastics production, most recycling systems worldwide cannot cope. In resource-constrained settings waste management is often limited to open burning which forms highly toxic, persistent dioxin compounds and particulate matter that imperil livelihoods, ecosystems and public health[24]. Multinational food and beverage companies bear major responsibility for these externalised costs of doing business[25].

## Implications of increased or decreased plastics use on food availability and quality

There are understandable concerns around potentially decreasing the use of plastics in specific food system applications in terms of the immediate effects this may have on food production, availability and quality. This may be because it is not entirely clear how alternate scenarios will play out for every application in different contexts. What is clear, however, is that plastics are already undermining Earth's vital support systems that underpin food production, availability and quality.

Global plastics production is projected to triple by 2060, and by 2040 will already account for 19% of GHGs[21,26]. Increases in plastic production are strongly associated with increases in plastic pollution, and will likely remain so even with the most optimistic upscaling in recycling and recovery[25,27,28]. Plastics and plastic chemicals currently in use and those already present in the environment ('legacy plastics') are already exacerbating impacts across all planetary boundaries—the fundamental systems upon which global food production, availability and quality are reliant[29–32]. These plastics will continue to fragment into micro- and nanoplastics and leach toxic chemicals, or absorb persistent organic pollutants, and contaminate the food chain[33].

In light of this, further increasing plastics production presents considerable risks for the future of food.

It is important to note that this plastic pollution comes from applications both inside and outside of the food system[34,35]. For example, as noted above, while agricultural plastics are major sources of soil pollution, they are not the only contributors. Others include fibres from clothes which contaminate sewage sludge used as fertilizer and tyre fragments from road run-off in close proximity to arable land. As such, decreasing plastic use only on farms without addressing wider systemic sources may not curtail impacts of plastics on agricultural production. Conversely, because any type of plastic production and pollution exacerbates pre-existing climate change impacts on food systems (some more than others), this contributes to vicious cycles which may necessitate greater use of plastics. For example, this may be the case where mulches or polytunnels are required in increasingly arid regions. These interactions and feedback loops illustrate the need for a whole food systems approach to ensure synergies with broader multi-sectoral responses to plastic pollution in all its forms.

To avoid regrettable substitutions and burden shifting, and to assess the consequences of increasing or decreasing food system plastics it will be important to apply essential use criteria (i.e. if the application is necessary for the health, safety, or the functioning of society, and if safe and sustainable alternatives are not currently available).

## Missing in food systems agendas?

Despite the evidence, recognition of food-related plastics as a system level challenge akin to climate change remains lacking in major 'food system transformation' agendas or debates. For instance, plastics gained little attention throughout the 2021 UN Food Systems Summit despite calls from plastics researchers. Similarly, the 2024 Global Policy Report on The Economics of the Food System Transformation[36] includes just one suggestion: to increase the use of 'bioplastics'. However, 'bioplastics' are an ambiguous, heterogeneous and questionable group of materials with both potential benefits and harms, and currently subject to scientific caution[37,38]. Plastics researchers recommend that the term should be avoided as it causes confusion about whether the plastic contains biodegradable properties/can be fully mineralised, bio-based, or both[39].

Is a lack or harmonisation of data limiting the discourse around plastics at the food systems level? Or, is a lack of discourse at the systems level limiting the generation and harmonisation of data? Or both? Some clues may be found in high profile food systems initiatives.

The Food Systems Countdown Initiative (FSCI)[40], compiled by a large, multi-disciplinary group of global experts is a timely 'indicator framework and holistic monitoring architecture to track food system transformation towards global development, health and sustainability goals.' Despite considerable evidence of direct and indirect implications across all Food Systems Countdown Initiative domains, plastics and their hazardous chemicals do not feature in the framework, because, as the authors note: "indicators of solid waste and chemical pollution attributable to food systems are wanting"[40]. Closely linked to the Food Systems Countdown Initiative is the Food Systems Dashboard[41] which allows users to explore data relating to

healthy and sustainable food systems. Of the 200+ indicators, none are specific to plastics.

While these examples suggest that plastics are not well-tracked in high level food systems surveillance, it is an oversimplification to suggest that they are altogether absent. Indeed, *some* elements of *some* food system plastics exist within established indicators. For instance, the indicator attributing a third of global GHGs to food systems[42] incorporates fuel chain, chemical inputs and incineration of plastic packaging but does not capture the full GHG burden of many other plastics used in agriculture, fisheries, storage and distribution. The analysis behind this indicator also states that "organic biomass fraction in solid waste is predominantly associated with food systems, while the non-organic fraction is not predominantly associated with food systems"[42]. This excludes the contribution of plastics to global waste streams, where for instance, in the USA, food and its packaging is estimated to account for over 45% of solid waste[43]. Nor does it capture the context-specific (mis) management of food-related plastic waste across different geographies that drives large variations in end-of-life emissions and onward impacts.

### Harmonising evidence for a fuller picture of food system plastics

Such a near absence of plastics in global food system discourse and indicator frameworks limits society's attempts to assess the extent to which food systems are (un)healthy, (un)safe, (un)sustainable and (in)equitable. It also limits how the magnitude of the plastics problem is understood and recognised throughout systems-level agenda-setting and discourse. Consequently, policies and programmes may be misinformed with potentially catastrophic unintended consequences. In some instances, this absence is due to a lack of adequate data to support monitoring and decision-making, however, some emerging data and indicators could be accelerated for this purpose (See Box 1: Aspirational indicators for future monitoring of food systems plastics).

Crucial research on specific food system plastics is accumulating, particularly around agricultural and packaging applications, including life cycle assessments[44] and hazardous plastic chemicals[10]. However, research agendas remain siloed among disciplines and/or food system sub-sectors; while data remain skewed towards particular geographies or outcomes[44]. Harmonised and interdisciplinary agenda-setting under a food systems framework could offer several key benefits:

Firstly, interdisciplinary research collaboration would collectively identify existing and persisting data gaps. This is important because currently there is no centralised database tracking specific polymers, chemicals, applications, fates and impacts of food system plastics in different contexts. Similarly, our collective understanding of the lived experiences and other socio-cultural factors that determine decision-making processes relating to different plastics used throughout the food system is lacking and would benefit from enhanced collaboration and recognition of different forms of knowledge[45].

Secondly, identifying data gaps will strengthen calls for transparency (including data disclosure, traceability and trackability) and accountability among food system actors driving plastics production. By the time plastics and their associated chemicals pollute (agro)ecosystems and contaminate food and humans, their origins are difficult (if not impossible) to trace. This hampers operationalising the polluter pays principle, extended producer responsibility and circular food system aspirations, as recognised by the Food Systems Countdown Initiative authors who "acknowledge the data gap in material pollution attributable to food systems"[40].

Thirdly, filling data gaps with robust evidence and knowledge will enable the tracking and tracing of multiple pathways by which food system plastics impact human and planetary health throughout their life cycles. In turn, this will substantiate robust essentiality criteria[46] needed to determine whether context specific plastic applications or functions are truly necessary for the health, safety and/or functioning of society, and whether safe and sustainable alternatives are currently available through a just transition.

### Box 1 | Aspirational indicators for future monitoring of food systems plastics

Examples of aspirational indicators for which data systems could be developed to monitor food system plastics:

Plastic polymers and chemicals of concern
- Virgin plastics across food supply chains
- Recycled plastics across food supply chains
- Plastic chemicals of concern across food system plastics

Land-use
- Land exposed to agricultural plastics

Trade and waste
- Food-related plastics and plastic waste import/exports
- Plastic waste from food supply chains

Environmental and human health impacts/costs
- GHG emissions of full lifecycle of plastics used across food supply chains
- Disease burden and costs of plastics (and plastic chemicals) attributable to food supply chains
- Ecological, social and economic costs of plastic pollution attributable to food supply chains

### The Global Plastics Treaty

The Global Plastics Treaty has been referred to as the most significant multilateral environmental agreement since the Paris Accord. More than just a global environmental treaty, its implications for food systems and human health are fundamental since it has the potential to affect plastic use across food production, processing, transportation, marketing, consumption, disposal, and removal and remediation. "Food" is mentioned 17 times in relation to food safety, food security and sustainability in the draft compilation text following the fourth round of negotiations[1] and has been frequently invoked by member state delegates, signifying just how central, contested and complex this issue is.

Groundswell around the negotiations has already driven debate and pressure around accountability and transparency, with over 3000 businesses, including many food companies, pledging to disclose plastic production and use data, albeit narrowly defined, via the Carbon Disclosure Project[47]. Similarly, linked to the negotiations, FAO has undertaken consultations towards a voluntary code of conduct on sustainable use of agricultural plastics[48]. While these are small steps, they indicate that momentum and magnitude of the issue is moving constituents into action. The challenge will be to knit such efforts together into a coherent body of interoperable data that can guide robust (real world and modelling) studies to present clear paths towards eliminating all but essential food system plastics, based on the precautionary principle[49,50].

Following ratification of the Global Plastics Treaty, national and regional policy implementation will follow, including obligations and compliance measures for member states, industries, and sectors. However, the effectiveness of the treaty remains in question with vested interests persistently seeking to weaken it e.g. through narrowing its scope and control measures, and limiting the means of reaching an agreement. Among these are hundreds of industry representatives, many with petrochemical, plastics, packaging, and commercial food system ties, who have flooded the treaty negotiations[51]. While it is legitimate for major drivers of the plastic problem to be consulted on certain issues, we echo calls for the treaty negotiations and other agenda-setting processes, including those directly focused on food systems, to be protected against conflicts of interest[52,53]. Furthermore, we support calls for an independent scientific subsidiary body of the future instrument, including Conflict of interest-free food systems scientists[54].

As independent scientists collaborating across disciplines and geographies to ensure Global Plastics Treaty negotiations are informed by robust independent evidence, we call on the food systems community to recognise plastics chemicals, polymers, products, alternatives and substitutes, technologies, systems and services as an integrated and multiscalar (global, regional, national) whole systems challenge for food. Elevating plastics in food systems discourse and agenda-setting will enable collaboration toward effective solutions that centre people and planet.

## Data availability
No data or code were used in this article.

## Code availability
No data or code were used in this article.

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

## Acknowledgements

The coordination of this paper was funded through UK Aid from the UK Government and the Bill & Melinda Gates Foundation via the Innovative Methods and Metrics for Agriculture and Nutrition Actions (IMMANA) programme, based at the London School of Hygiene & Tropical Medicine. Authors would like to thank the reviewers for their valuable inputs and suggestions.

## Author contributions

All authors (J.Y., M.D., J.M., B.C.A., M.F.D., A.C.C., W.C.J., S.K., E.K., P.S., M.W., T.F.) contributed to the conceptualisation, writing, editing and reviewing of this manuscript. J.Y. led the drafting rounds.

## Competing interests

The authors declare no competing interests.
