## [Transparent Peer Review file · Communications Earth & Environment]

Plastics matter in the food system

Corresponding Author: Mr Joe Yates

This manuscript has been previously reviewed at another journal that is not operating a transparent peer review scheme. The manuscript was considered suitable for publication without further review at Communications Earth & Environment.

Version 0:

Decision Letter:

Dear Mr Yates,

Thank you for submitting your revised manuscript titled "Plastics matter in the food system". We have discussed your manuscript within our editorial team, and we are delighted to say that we are happy, in principle, to publish a suitably revised version in Communications Earth & Environment, under the condition that you provide a discussion of how increased or decreased plastic use affects food availability and quality.

We therefore invite you to revise your paper one last time to address the remaining concerns. At the same time we ask that you edit your manuscript to comply with our format requirements and to maximise the accessibility and therefore the impact of your work.

EDITORIAL REQUESTS:

****Please take care to match our formatting and policy requirements. We will check revised manuscript and return manuscripts that do not comply. Such requests will lead to delays. ****

SUBMISSION INFORMATION:

OPEN ACCESS:

Communications Earth & Environment is a fully open access journal. Articles are made freely accessible on publication. For further information about article processing charges, open access funding, and advice and support from Nature Research, please visit <https://www.nature.com/commsenv/open-access>

Link Redacted

Best regards,

Martina Grecequet, PhD
Senior Editor,
Communications Earth & Environment
@CommsEarth
